:✪: PLOS | ONE

# KLF4 is required for suppression of histamine synthesis by polyamines during bone marrow-derived mast cell differentiation

Kazuhiro Nishimura●[1]*, Moemi Okamoto[1], Rina Shibue[1], Toshio Mizuta[1], Toru Shibayama[1], Tetsuhiko Yoshino[1], Teruki Murakami[1], Masashi Yamaguchi[2], Satoshi Tanaka[3], Toshihiko Toida[1], Kazuei Igarashi[1,4]

**1** Graduate School of Pharmaceutical Sciences, Chiba University, Chiba, Japan, **2** Medical Mycology Research Center, Chiba University, Chiba, Japan, **3** Department of Pharmacology, Division of Pathological Sciences, Kyoto Pharmaceutical University, Kyoto, Japan, **4** Amine Pharma Research Institutes, Chiba, Japan

* kaznishi@faculty.chiba-u.jp

## Abstract

Mast cells have secretory granules containing chemical mediators such as histamine and play important roles in the immune system. Polyamines are essential factors for cellular processes such as gene expression and translation. It has been reported that secretory granules contain both histamine and polyamines, which have similar chemical structures and are produced from the metabolism of cationic amino acids. We investigated the effect of polyamine depletion on mast cells using bone marrow-derived mast cells (BMMCs). Polyamine depletion was induced using α-difluoromethylornithine (DFMO), an irreversible inhibitor of ornithine decarboxylase. DFMO treatment resulted in a significant reduction of cell number and abnormal secretory granules in BMMCs. Moreover, the cells showed a 2.3-fold increase in intracellular histamine and up-regulation of histidine decarboxylase (HDC) at the transcriptional level during BMMC differentiation. Levels of the transcription factor kruppel-like factor 4 (KLF4) greatly decreased upon DFMO treatment; however, *Klf4* mRNA was expressed at levels similar to controls. We determined the translational regulation of KLF4 using reporter genes encoding *Klf4-luc2* fusion mRNA, for transfecting NIH3T3 cells, and performed *in vitro* translation. We found that the efficiency of KLF4 synthesis in response to DFMO treatment was enhanced by the existence of a GC-rich 5′-untranslated region (5′-UTR) on *Klf4* mRNA, regardless of the recognition of the initiation codon. Taken together, these results indicate that the enhancement of histamine synthesis by DFMO depends on the up-regulation of *Hdc* expression, achieved by removal of transcriptional suppression of KLF4, during differentiation.

## Introduction

Polyamines are small basic molecules with multiple amino groups and are involved in cell proliferation and differentiation [1, 2]. Three polyamines (putrescine, spermidine and spermine)

**Data Availability Statement:** All relevant data are within the manuscript and its Supporting Information files.

**Funding:** This work was supported by a Grant-in-Aid for Scientific Research 15K07921 from the Ministry of Education, Culture, Sports, Science and Technology, Japan.

**Competing interests:** The authors have declared that no competing interests exist.

are present in mammalian cells at the millimolar level, and are stringently controlled by biosynthesis, degradation, and transport [3]. Putrescine is synthesized from ornithine by ornithine decarboxylase (ODC), and spermidine is synthesized from putrescine by the addition of an aminopropyl group donated from decarboxylated *S*-adenosylmethionine, which is synthesized from *S*-adenosylmethionine by *S*-adenosylmethionine decarboxylase (AdoMetDC). Similarly, spermine is synthesized from spermidine by the addition of another aminopropyl group. These enzymes catalyze rate-limiting steps in polyamine biosynthesis. Studies on mice with disruption of ODC- and AdocMetDC-encoding genes revealed that these deficient mice were embryonic lethal [4, 5]. It has been demonstrated that cellular polyamines mostly interact with RNA and consequently affect translation efficiency of several kinds of proteins [6]. The mRNAs encoding these proteins have characteristic nucleotide sequences in their 5′-UTRs that are difficult to translate. In addition, the term 'polyamine modulon', is used to describe genes encoding proteins regulated by polyamines at the translational level [7]. We have previously reported three kinds of genes (*Cct2*, *Hnrpl*, and *Pgam1*) as the polyamine modulon and clarified one of the mechanisms of polyamine stimulate protein synthesis in eukaryotes [8]. Thus, polyamine mediated stimulation of protein synthesis contributes to cell proliferation and differentiation.

Mast cells are present in connective and submucosal tissues of mammals and are known to be involved in immunological processes contributing to both innate and adaptive immunity in defense against pathogens [9, 10]. They contain many secretory granules with serglycin proteoglycans (PGs) and important chemical mediators [11]. Cutaneous mast cells preferentially express serglycin PGs with heparin chains in humans and mice, whereas jejunal mast cells and BMMCs preferentially express serglycin PGs with chondroitin sulfate-E chains [12]. The absence of serglycin was shown to result in absence of metachromatic staining and resulted in severely defective storage of granule proteases and biogenic amines [13, 14]. Therefore, serglycin PGs manifest strong electronic interactions with secretory granule compounds and play an essential role in maintaining the morphological characteristics of mast cells [11].

It was previously found that blood coagulation in the presence of heparin was enhanced by polyamines due to interactions between heparin and polyamines [15]. Moreover, glycosaminoglycan has been shown to be involved in polyamine transport: heparan sulfate on glypican-1 acted as a vehicle for polyamine uptake [16]. García-Faroldi *et al* reported that polyamine depletion resulted in the upregulation of *Hdc* expression and activity, accompanied by increased histamine levels during early stages of BMMC differentiation [17]. However, the mechanism of upregulation of *Hdc* expression was unclear. A study in 2004 showed that KLF4 suppresses expression of *Hdc* in gastric cancer [18]. There is little known about the role of KLF4 in mast cells.

Here, we investigated the effects of polyamine depletion, using a low concentration of DFMO, on BMMC differentiation and function. We demonstrated that KLF4 synthesis is regulated at the translational level by polyamines and is involved in histamine synthesis in mast cells.

## Materials and methods

### Animals

All animal experiments were approved by the Institutional Animal Care and Use Committee of Chiba University and carried out according to the Guidelines for Animal Research of Chiba University. Female C57BL/6 mice were obtained from Japan SLC, Inc.

## Materials

Calcium ionophore A23187 was purchased from Sigma-Aldrich. Purified mouse IgE κ Isotype Control (clone C38-2) and purified rat anti-mouse IgE (clone R35-72) were from BD Biosciences Pharmingen. Recombinant murine IL-3 was purchased from PeproTech, Inc.

**Preparation of BMMCs.** BMMCs wer prepared as previously described [19]. Female, 8–12 week-old, C57BL/6 mice were euthanized by cervical dislocation, during isoflurane anesthesia, tibia bone marrow cells were isolated, and cultured in RPMI-1640 containing 10% fetal bovine serum, 10 ng/mL IL-3, 50 μM β-mercaptoethanol, 0.1 mM non-essential amino acids, 50 U/mL penicillin G and 50 U/mL streptomycin. Cells were subcultured at $5 \times 10^5$ cells/mL with 3–4 d intervals for 28–34 d to obtain mature BMMCs. Maturation of BMMCs was confirmed by toluidine blue staining.

**Transmission electron microscopy.** Cells were fixed with 2.5% glutaraldehyde and kept on ice. They were post-fixed with 1% osmium tetroxide, dehydrated with a graded series of ethanol, and embedded in an epoxy resin [20]. Ultrathin sections were cut, stained with uranyl acetate and lead citrate, and observed under a JEM-1400 electron microscope (JEOL Ltd.) [21].

**Degranulation assay.** Cells ($1 \times 10^6$) were stimulated by either the calcium ionophore A23187 or IgE/anti-IgE for this assay. In the A23187 stimulation method, the cells were washed and suspended in Tyrode's buffer (10 mM HEPES/NaOH (pH 7.4), 130 mM NaCl, 5 mM KCl, 1.4 mM CaCl$_2$, 1 mM MgCl$_2$, 5.6 mM glucose, and 0.1% BSA). After preincubation for 10 min at 37 ˚C, the cells in 200 μL of Tyrode's buffer containing 2 μM A23187 were incubated for 15 min at 37 ˚C and placed on ice for 10 min. In the IgE/anti-IgE stimulation method, the cells were washed with phosphate buffer saline (PBS), suspended with 1 μg/mL anti-TNP mouse IgE in conditioned medium and incubated for sensitization for 6 h at 37 ˚C. The sensitized cells were washed with PBS, suspended in 200 μL of Tyrode's buffer containing 2 μg/mL anti-mouse IgE, incubated for 1 h at 37 ˚C and placed on ice for 10 min. To evaluate the degranulation rate of the two methods, β-hexosaminidase activity in supernatants and cell pellets (solubilized with 1% Triton-X-100 in Tyrode's buffer) was determined. The samples, with 1 mM *p*-nitrophenyl *N*-acetyl-β-D-glucosaminide in 50 mM citrate buffer (pH 4.5), were incubated for 1 h at 37 ˚C; 2 volumes of 0.1 M Na$_2$CO$_3$/NaHCO$_3$ (pH 10) was added to stop the reaction and the release of *p*-nitrophenol was measured spectrophotometrically at 405 nm. The degranulation rate was calculated using the absorbance of the supernatant/the total absorbance (supernatant + cell pellet).

**Measurement of amines.** Intracellular amine (polyamines and histamine) levels in BMMCs were determined according to the method described previously [22] with minor modifications. Cells ($1 \times 10^6$) or culture medium were treated with final 5% trichloroacetic acid and centrifuged at 10,000 x g for 5 min. Isocratic single or tandem-mode HPLC system, with one or two columns (TSKgel polyaminepak; Tosoh Corporation) were used to analyze the supernatants. Eluent (pH 5.6) was used to separate amines for 30 min in the single-mode, and eluent (pH 5.3) for 120 min in the tandem-mode. The amount of protein present in trichloroacetic acid -precipitates was evaluated using Bio-Rad Protein Assay (Bio-Rad Laboratories).

**Measurement of HDC activity.** HDC assay was performed following the method described [23], with minor modifications. Cells ($5 \times 10^6$) were washed with PBS and suspended with 320 μL of HDC assay buffer (100 mM potassium phosphate (pH 6.8), 0.2 mM dithiothreitol, 10 μM pyridoxal 5′-phosphate, 2% polyethylene glycol 300, 20 μM FUT-175, and 1% TritonX-100). Thereafter 150 μL of cell lysate was centrifuged at 16,000 x g for 15 min, incubated with 50 μL of HDC buffer containing 3.2 mM L-histidine, for 2 h, and treated with 5%

trichloroacetic acid (final concentration) for stopping the enzymatic reaction. Histamines produced were analyzed by the single-mode HPLC system described above.

**Quantitative RT-PCR analysis.** Total RNA was isolated from $1 \times 10^6$ cells using the RNeasy Micro Kit (Qiagen) and used for first-strand cDNA synthesis using SuperScript™ II Reverse Transcriptase (Invitrogen). Real-time PCR was performed using KAPA SYBR FAST® qPCR kit (Kapa Biosystems Inc.) and Eco Real-Time PCR System (Illumina Inc.) The primer pairs (forward, reverse) used for PCR were as follows; *Hdc*, 5′-ATC TCT GGT CAG AAG CGA CCC TTC-3′, 5′-GCT CCT GGC TGC TTG ATG ATC TTC-3′, *Klf4*, 5′-GCA GGC TGT GGC AAA ACC TAT AC-3′, 5′-CTG ACT TGC TGG GAA CTT GAC-3′, *Gapdh*, 5′-GGT ATC GTG GAA GGA CTC ATG AC-3′, 5′-ATG CCA GTG AGC TTC CCG TTC AGC-3′.

**Western blotting.** Western blot analysis was performed as described previously [24]. Primary antibodies used were rabbit polyclonal KLF4 antibody (1:2,000; GTX101508; GeneTex) and mouse monoclonal β-actin antibody (1:2,000; ab8226; Abcam). Secondary antibodies used were rabbit IgG HRP Linked Whole Ab (1:10,000; NA943; GE Healthcare) and mouse IgG HRP Linked Whole Ab (1:10,000; NA941; GE Healthcare).

**Reporter plasmid construction.** The pGL4.13[*luc2*/SV40] vector was purchased from Promega. For the ΔE2 plasmid, the *Klf4* E1 fragment (605 bp) was obtained by PCR using primers (forward, 5′-CAA AAA GCT TAG TTC CCC GGC CAA GAG AGC GAG-3′, reverse, 5′-GAT CAA GCT TAA TGT GGG GGC CCA GAA GG-3′), digested with HindIII and inserted into compatible sites on pGL4.13. For the WT plasmid, the partial *Klf4* E1-E2 fragment (179 bp) was obtained by PCR using the primers (forward, 5′-CAA AAA GCT TAG TTC CCC GGC CAA GAG AGC GAG-3′, reverse, 5′-CGC TGG GCC CTT CTT AAT GTT TTT GGC ATC TTC CAT AGA CTC GCC AG-3′), digested with ApaI, and inserted into compatible sites on the ΔE2 plasmid. For the ΔE1 plasmid, the WT plasmid was digested with HindIII, to delete the E1 fragment (660 bp), and self-ligated. Three mutant plasmids (uORFmut, 1st ATGmut, and 2nd ATGmut) were generated by making a single nucleotide substitution (A→G) in the WT plasmid. Mutant strands were synthesized by PCR using primers (forward, reverse): uORFmut, 5′-CGT GAC CCG CGC CCG TGG CCG CGC GCA CCC-3′, 5′-GGG TGC GCG CGG CCA CGG GCG CGG GTC ACG-3′; 1st ATGmut, 5′-GGC CCC CAC ATT AGT GAG GTA GGT GAG ATG-3′, 5′-CAT CTC ACC TAC CTC ACT AAT GTG GGG GCC-3′; 2nd ATGmut, 5′-CCT GGC GAG TCT GAC GTG GAA GAT GCC AAA AAC-3′, 5′-GTT TTT GGC ATC TTC CAC GTC AGA CTC GCC AGG-3′, digested with DpnI and transformed in *Escherichia coli*. Mutations were confirmed by DNA sequencing.

**Transfection and reporter assay.** Reporter plasmids were transfected into NIH3T3 cells using polyethylenimine (PEI), as previously described [25]. The cells were maintained in DMEM, containing 10% fetal bovine serum, 50 U/mL penicillin G, and 50 U/mL streptomycin. Cells ($6 \times 10^4$) were plated on 12-well plates for 12 h, cultured in the absence or presence of 5 mM DFMO for 12 h, and replaced with 250 μL of Opti-MEMI before transfection. To form PEI/DNA polyplexes, 3 μg of PEI, 0.5 μg of reporter plasmid, and 0.05 μg of pGL4.75 [*hRluc*/CMV] (used as an internal control for transfection efficiency) were mixed in 50 μL of lactate-buffered saline (20 mM sodium lactate (pH 4.0), 150 mM NaCl) and added to each well. After 8 h, the cells were further cultured with DMEM in the absence or presence of 5 mM DFMO for 16 h. Luciferase activities were measured using the Dual-luciferase Reporter Assay System (Promega) with luminometer GL-210A (Microtec Co., Ltd.), and firefly luciferase activity of reporter plasmid was normalized with *Renilla* luciferase activity.

**In vitro translation.** To prepare *luc2* fused with the T3 promoter for *in vitro* transcription, T3-*luc2* and T3-*Klf4-luc2* templates were obtained by PCR using the pGL4.13 and ΔE2

plasmids as template and the following primers (forward, reverse); 5′-GAA ATA TTA ACC CTC ACT AAA GGG CTT TTG CAA AAA GCT T-3′, 5′-TGT TAA CTT GTT TAT TGC AG-3′. The *luc2* and *Klf4-luc2* mRNAs were synthesized by using mMESSAGE mMA-CHINE™ T3 Transcription Kit (Invitrogen), according to the manufacturer's protocol. FM3A nuclease-treated cell lysates, for *in vitro* translation in FM3A cell-free systems, was prepared as described previously [26]. The reaction mixture (10 µL) contained 15 mM Hepes-KOH (pH 7.6), 10 µM hemin, 75 mM potassium acetate, 2 mM dithiothreitol, 0.5 mM glucose 6-phosphate, 1 mM ATP, 0.4 mM GTP, 8 mM creatine phosphate, 150 µg/mL of creatine kinase, 10 µM FUT-175, 30 µM of each of the 20 amino acids, 1x protease inhibitor cocktail, 1.5 mM magnesium acetate, 0.5 µg of synthesized mRNA, different concentrations of polyamines (spermidine or spermine), and 0.1 mg of FM3A nuclease-treated cell lysate. After incubation at 37 ˚C for 1 h, the synthesized proteins were detected by measuring luciferase activity, using the Luciferase Assay System (Promega).

**Statistical analysis.** The data were analyzed for statistical significance using the two-tailed student's t-test. $P < 0.05$ was considered as statistically significant.

## Results

### Effect of low concentration-DFMO treatment during BMMC differentiation

To evaluate the effect of DFMO treatment during BMMC differentiation, bone marrow cells were cultured in the absence or presence of 5 mM DFMO, as shown in previous reports [17, 27], however, all cells died before maturation. We then modified the culture conditions and grew the cells in the absence or presence of 0.1 mM DFMO. The cells were subcultured for four weeks under this condition. As shown in Fig 1A, relative cell numbers during differentiation significantly decreased at a low concentration of DFMO, and no dead cells appeared to suggest apoptosis. To confirm the correlation between decrease in cell number and cellular polyamine concentration, the polyamine levels were analyzed at several time points during differentiation. Levels of putrescine and spermidine were efficiently reduced during differentiation, and spermine levels increased 2-fold only on day 28 (Fig 2A). A previous study reported altered ultrastructure of secretary granules in DFMO-treated BMMCs [27]. Therefore, we performed transmission electron microscopy analysis. Control BMMCs showed a distinctive dense core structure in secretary granules (Fig 1B and 1C), however, DFMO-treated cells were filled with amorphous structures (Fig 1D and 1E). We assayed degranulation by using two independent stimuli; the calcium ionophore, A23187 or IgE/anti-IgE. The degranulation rates of BMMCs were unaffected by DFMO treatment (Fig 1F). These results indicated that the effect of 0.1 mM DFMO treatment during BMMC differentiation was similar to that of 5 mM DFMO treatment.

### Increase of intracellular histamine in DFMO-treated BMMCs

To determine the effects of DFMO treatment on intracellular histamine levels during BMMC differentiation, we performed simultaneous analysis of polyamines and histamine. Although intracellular histamine levels were similar in response to DFMO treatment during differentiation, until day 21, there was a 2.3-fold increase on day 28 (Fig 2B).

We then investigated whether histamine levels were affected by polyamine levels after BMMC differentiation. As shown in Fig 3, BMMCs in the absence or presence of DFMO, were placed in four dishes on day 28 and further cultured until day 34. Reduced putrescine and spermidine levels, in the presence of DFMO, were recovered to control levels by changing the

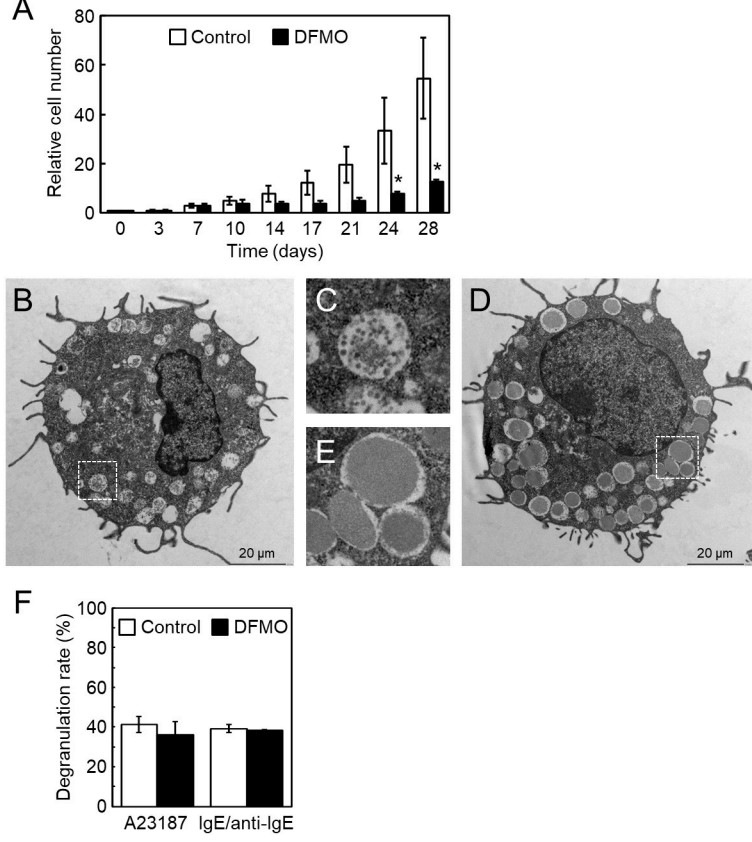

**Fig 1. Cell number and morphology of BMMCs in the absence or presence of DFMO.** (A) Starting with 0.1 mM DFMO treatment from day 3, the cells were cultured until day 28. The relative cell number on day 0 is shown as 1. Standard deviation (s.d.) was calculated; n = 3 replicates. *p < 0.05 compared with control. (B) Control and (D) DFMO-treated cells on day 28, observed with transmission electron microscopy. (C) and (E) Boxes with white dots from (B) and (D) in larger magnification, respectively. (F) β-hexosaminidase activity of supernatant and cell pellet after degranulation stimuli (calcium ionophore A23187 or IgE/anti-IgE) was given on day 28. Degranulation rate (%) ± s.d. is shown as [supernatant]/[supernatant + cell pellet], n = 3 replicates.

culture condition (without DFMO) on day 34, and spermine levels were slightly decreased. Under this condition, histamine levels were still high. DFMO treatment from day 28 did not affect histamine levels. These results indicated that lower levels of putrescine and spermidine may be required to store higher levels of histamine during BMMC differentiation, but not after their differentiation.

## Histidine decarboxylase activity is regulated by KLF4 during BMMC differentiation

To determine whether higher histamine levels, produced due to DFMO treatment, correlated with HDC activity, cell lysates were prepared at different time points during BMMC differentiation and used for HDC assay. As shown in Fig 4A, activity of DFMO-treated cells between day 14 and 21 were 2-fold higher than control cells. We investigated the expression levels of *Hdc* mRNA on day 7, 14, and 21 by qRT-PCR. *Hdc* mRNA levels of DFMO-treated cells between day 14 and 21 were significantly higher, compared to control cells (Fig 4B). It has been previously reported that the transcription factor, KLF4, suppresses expression of *Hdc* [18]. We hypothesized that the decrease of KLF4 level by DFMO treatment increased *Hdc*

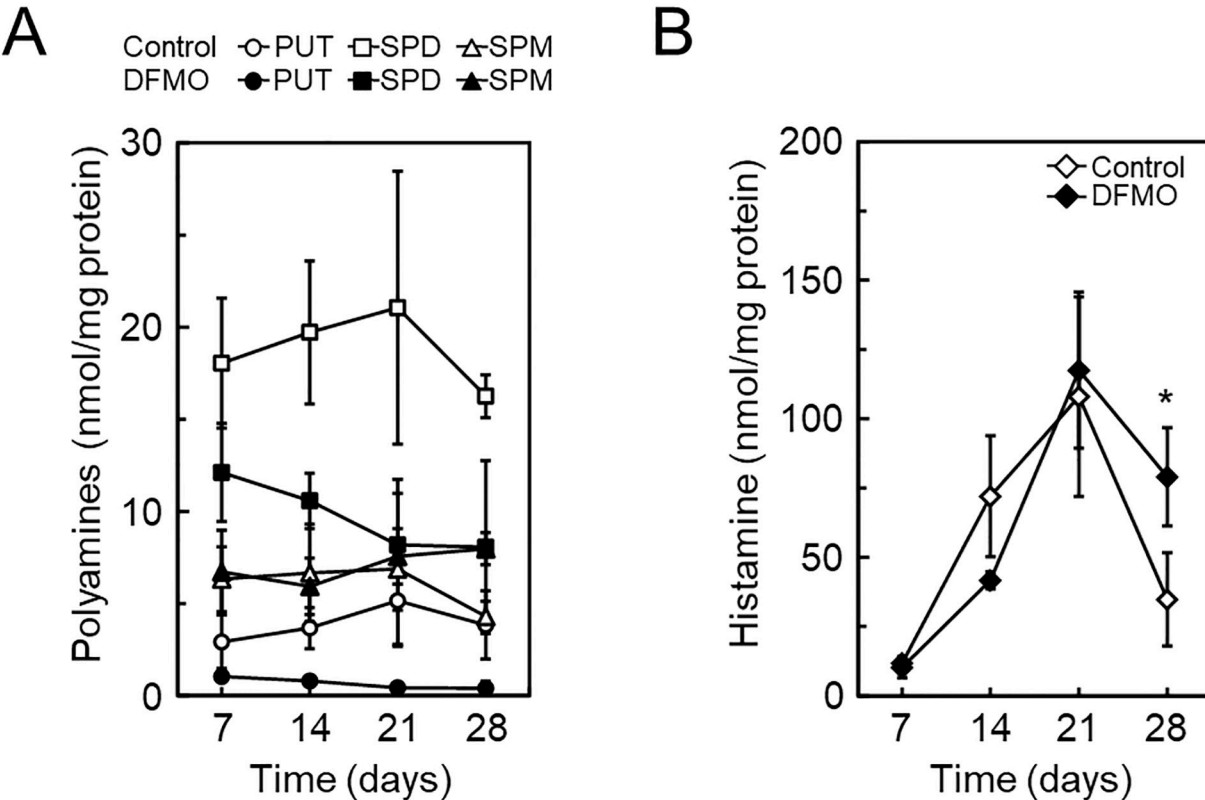

**Fig 2. Polyamine and histamine levels during BMMC differentiation in the absence or presence of DFMO.** During BMMC differentiation, cells on day 7, 14, 21, and 28 were used for analysis of intracellular polyamine (A) and histamine (B) content by HPLC. Mean ± s.d. is shown, n = 3 replicates. *p < 0.05, compared with control.

mRNA level. To confirm this hypothesis, we determined the amount of KLF4 (483 amino acids) during differentiation by western blotting. As shown in Fig 4C, S1 and S2 Figs, the amount of KLF4 in DFMO-treated cells, on day 14 and 21, was lower compared to control cells. qRT-PCR was further performed to determine *Klf4* mRNA levels on day 7, 14, and 21; *Klf4* mRNA levels did not decrease in response to DFMO treatment (Fig 4D). The *in vivo* binding of KLF4 on the *Hdc* promoter was examined by ChIP assay; a fragment of the proximal promoter region was amplified using anti-KLF4 precipitated-DNA (S3 Fig). Moreover, to exclude non-specific effects of DFMO, we carried out a recovery experiment by adding 100 μM putrescine. During differentiation, cell number and intracellular amines recovered to levels similar to control cells (S4 Fig). Importantly, *Hdc* and *Klf4* mRNA and KLF4 levels also recovered by the addition of putrescine (Fig 4B–4D). These results suggested that *Klf4* expression is translationally regulated by polyamines, and decreased KLF4 due to reduced polyamines resulted in the upregulation of *Hdc* expression.

## Involvement of the 5′-UTR of Klf4 mRNA in the translational regulation by polyamines

We have named the genes encoding proteins whose syntheses are enhanced by polyamines at the translational level as the 'polyamine modulon' [7, 28, 29]. They have a characteristic nucleotide sequence and structure at the 5′-UTR end of the mRNA. Therefore, we investigated the

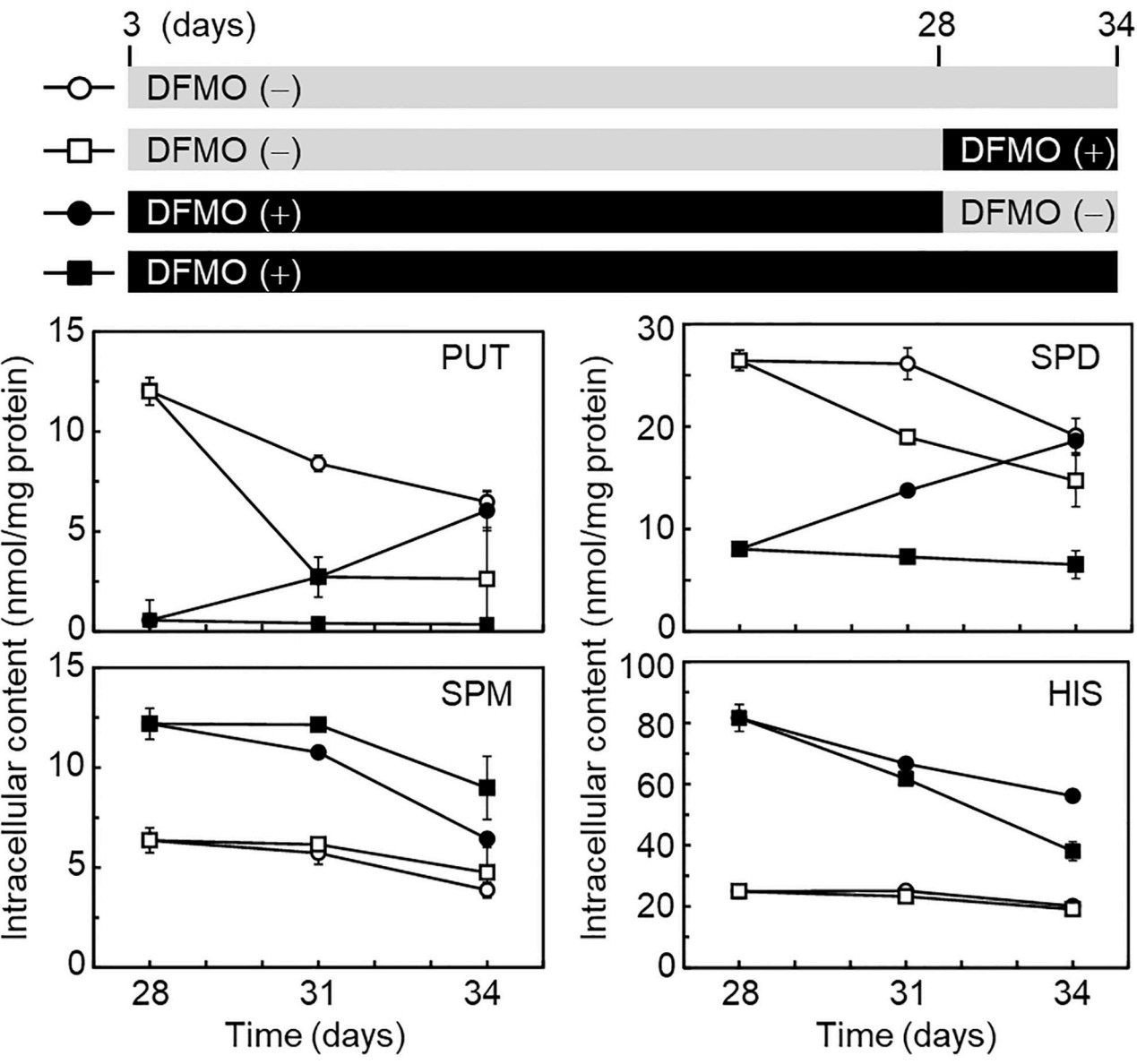

**Fig 3. Effect of DFMO on the level of polyamines and histamine after BMMC differentiation.** After 28 d of BMMC differentiation, in the absence or presence of DFMO, the cells were continuously cultured for 6 d under four different conditions, as shown in the upper panel. At each time point on day 28, 31, and 34, the level of polyamines and histamine was analyzed by HPLC. Mean ± s.d. is shown, n = 3 replicates.

nucleotide sequence of mouse *Klf4* (Gene ID: 16,600) in NCBI (https://www.ncbi.nlm.nih.gov/gene/). *Klf4* consists of 5 exons and 4 introns. There were two candidates for the initiation codon ATG in *Klf4*; the 1st and 2nd ATGs were located on exon1 (E1) and exon2 (E2), respectively (Fig 5A). The length of the 5′-UTR of *Klf4* mRNA is 604 nucleotides (nt) if translation starts from the 1st ATG. The GC content, from the transcriptional start site to 140 nt, on E1 was 81.4% (114/140). To investigate whether the 5′-UTR of *Klf4* mRNA is involved in translational regulation by polyamines, reporter plasmids were prepared and used for transfection into NIH3T3 cells (Fig 5A). In comparison with the pGL4.13 vector, relative luciferase activity of the ΔE1 plasmid was similar, however, the activities of WT and ΔE2 plasmids, containing the long E1 fragment, were decreased (left panel, Fig 5B). We tested the effect of DFMO

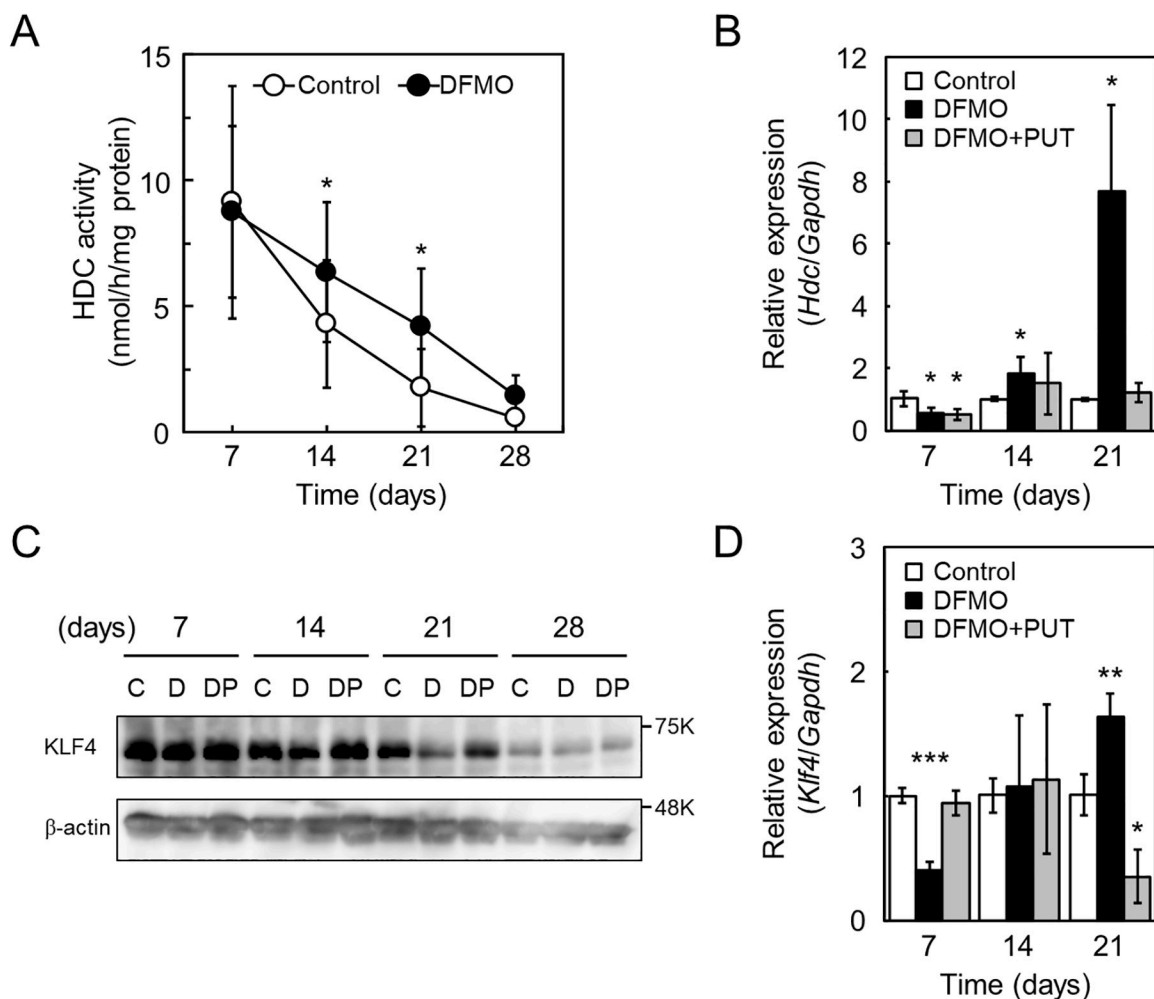

**Fig 4. *Hdc* and *Klf4* expression during BMMC differentiation in the absence or presence of DFMO.** (A) During BMMC differentiation, cells on day 7, 14, 21, and 28 were collected and used to prepare cell lysates. HDC assays were performed after dialysis of the cell lysate. Mean ± s.d. is shown, n = 3 replicates. *p < 0.05 compared with control. (B) Total RNA extracted from cells on day 7, 14, and 21 was isolated and used for qRT-PCR. Relative expression of *Hdc* mRNA was normalized to *Gapdh* and control expression is shown as 1. Mean ± s.d. is shown, n = 3 replicates. *p < 0.05 compared with control. (C) During BMMC differentiation, cells on day 7, 14, 21, and 28 were collected and analyzed for KLF4 and β-actin by western blotting. Lanes (C, D, and DP) represent control, DFMO, and DFMO + PUT, respectively. (D) Total RNA from cells on day 7, 14, and 21 was isolated and used for qRT-PCR. Relative expression of *Klf4* mRNA was normalized to *Gapdh*; control expression is shown as 1. Mean ± s.d. is shown, n = 3 replicates. *p < 0.05, **p < 0.01, and ***p < 0.001, compared with control.

treatment; values are presented as the ratio of relative luciferase activity of DFMO (+/−). The ratio of WT and ΔE2 plasmids was significantly reduced compared with the pGL4.13 vector, and that of the ΔE1 plasmid was equal (right panel, Fig 5B). As shown in Fig 5A, an upstream open reading frame (uORF, 180 nt) is present on E1 of *Klf4*. It is known that uORFs may negatively regulate translation of the downstream ORFs; therefore, uORF may be involved in translational regulation by polyamines. The uORFmut plasmid was prepared by a single nucleotide substitution (A→G) in the WT plasmid (uORFmut, Fig 5A). As shown in Fig 5C, relative luciferase activity of the uORFmut plasmid was 2-fold higher than the WT plasmid (left panel, Fig 5C); however, DFMO treatment resulted in no difference between WT and uORFmut (right panel, Fig 5C). Furthermore, we determined initiation codon usage efficiency by using 1st ATGmut and 2nd ATGmut plasmids (Fig 5A). The relative luciferase activity of 1st ATGmut

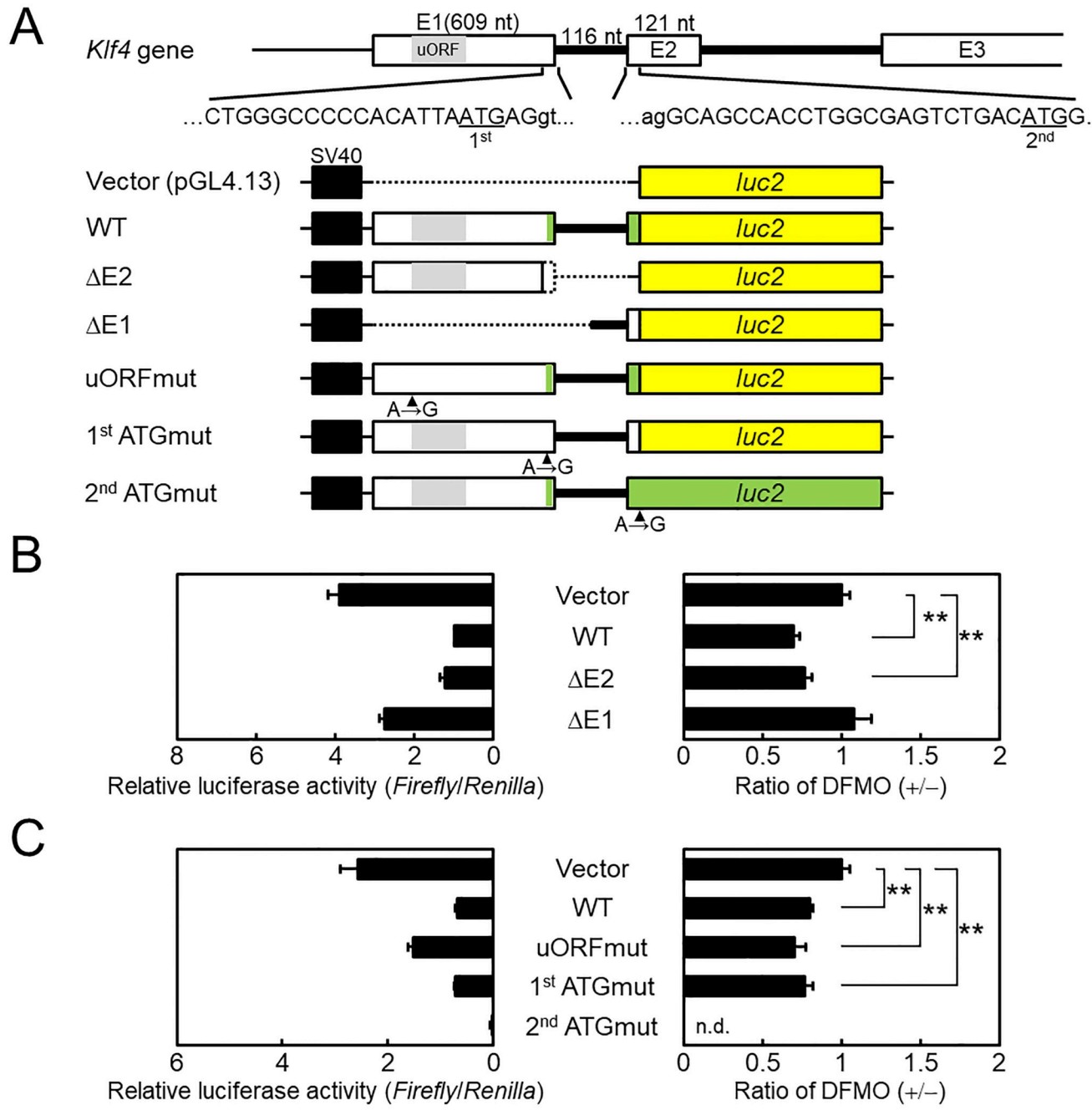

**Fig 5. Effect of polyamines on luciferase activity from normal and modified *Klf4-luc2* reporter plasmids in NIH3T3 cells.** (A) Schematic illustration of mouse *Klf4* (E1-E3) and various *Klf4-luc2* reporter plasmids are shown. Nucleotide sequences for exons and introns are represented as uppercase and lowercase, respectively. The two initiation codons (ATG) on *Klf4* are underlined as 1st and 2nd. The dotted line on *Klf4-luc2* reporter plasmids means that the region was deleted from WT plasmid. The site of mutated initiation codon, by site-directed mutagenesis, is indicated by an arrowhead (A→G). The green box in the plasmid illustration means the case of translation from 1st initiation codon on E1. The yellow box shows *luc2* ORF. (B) and (C); NIH3T3 cells co-transfected with the reporter plasmids and pGL4.75[*hRluc*/CMV] in the absence or presence of 5 mM DFMO, respectively. The graph on the left shows relative luciferase activities measured as the ratio of firefly and *Renilla* luciferase activity. The graph on the right shows data calculated as the ratio of relative luciferase activity of DFMO (+/−). The values of vector are shown as 1. Mean ± s.d. is shown, n = 3 replicates. **p < 0.01 compared with vector. n.d., not determined.

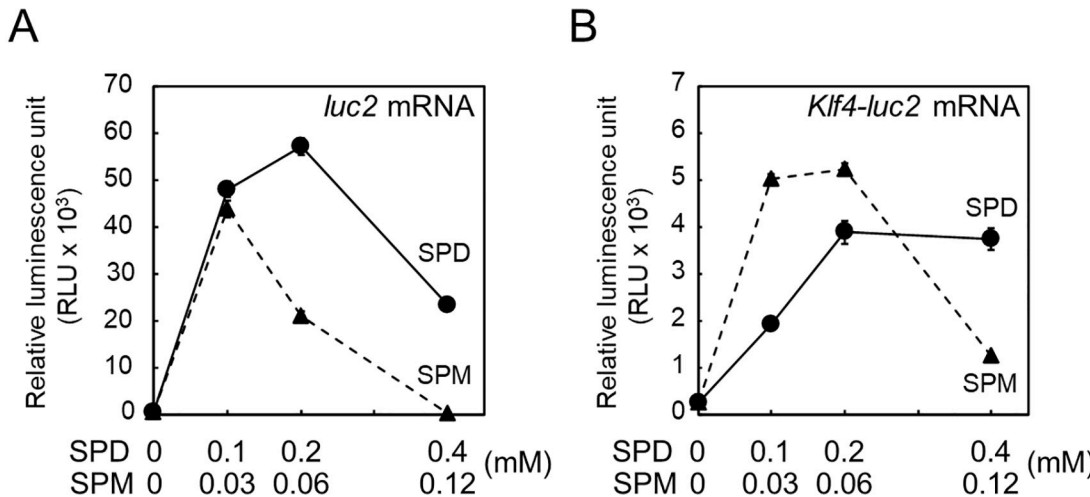

**Fig 6. Effect of polyamines on *Klf4-luc2* translation in an FM3A cell-free system.** *In vitro* translation of (A) *luc2* and (B) *Klf4-luc2* mRNAs were performed as described in MATERIALS AND METHODS. The level of luciferase synthesized from *luc2* was measured using Luciferase Assay System as relative luminescence unit (RLU). Mean ± s.d. is shown, *n* = 3 replicates.

plasmid was equal to the WT plasmid; however, the activity of the 2nd ATGmut plasmid almost ceased to be visible, regardless of DFMO treatment (Fig 5C). This result suggests that the 1st ATG does not operate as an initiation codon for KLF4. Taken together, we conclude that the 5'-UTR of *Klf4* mRNA is involved in translational regulation by polyamines, but the presence of uORF and two initiation codons is unrelated.

Finally, the effects of spermidine and spermine on translation of *Klf4-luc2* mRNA were examined using *in vitro* synthesized mRNAs in an FM3A cell-free system. The *Klf4-luc2* mRNA was synthesized from T3-*Klf4-luc2* of the ΔE2 plasmid (see Fig 5A) fused with a T3 promoter for *in vitro* transcription. Similarly, the *luc2* mRNA was synthesized from T3-*luc2* of the pGL4.13 vector as control. The translation of *luc2* mRNA occurred in the presence of 0.1 mM spermidine; however, translation of *Klf4-luc2* mRNA required a 2-fold higher spermidine concentration (circles and solid line, Fig 6A and 6B, respectively). On the other hand, the optimal concentration of spermine required for the stimulation of translation was 2-fold higher for *Klf4-luc2* mRNA (0.03 to 0.06 mM) compared to that for *luc2* mRNA (0.03 mM) (triangle and dashed line, Fig 6A and 6B, respectively). These results indicate that the 5′-UTR of *Klf4* mRNA influenced the sensitivity of polyamine concentration for translation.

## Discussion

Mast cells, derived from hematopoietic stem cells, are major players in IgE-mediated allergic reactions, and contain abundant chemical mediators such as histamine. Polyamines, which are essential for cell proliferation and differentiation, are abundant in mast cells. In this study, we demonstrated that intracellular histamine levels increased with an increase in *Hdc* expression and a decrease in KLF4 level during BMMC differentiation. Therefore, we propose a model that suggests that translational regulation of KLF4 by DFMO-induced polyamine depletion is required for the suppression of histamine synthesis during BMMC differentiation (Fig 7).

In 2009, García-Faroldi *et al* reported that DFMO-induced *Hdc* mRNA up-regulation was observed in early bone marrow cell cultures [17]. The reason that we could not reproduce BMMC differentiation under their DFMO treatment condition is unknown. However, low DFMO concentration treatment showed a similar effect, such as morphological change, on

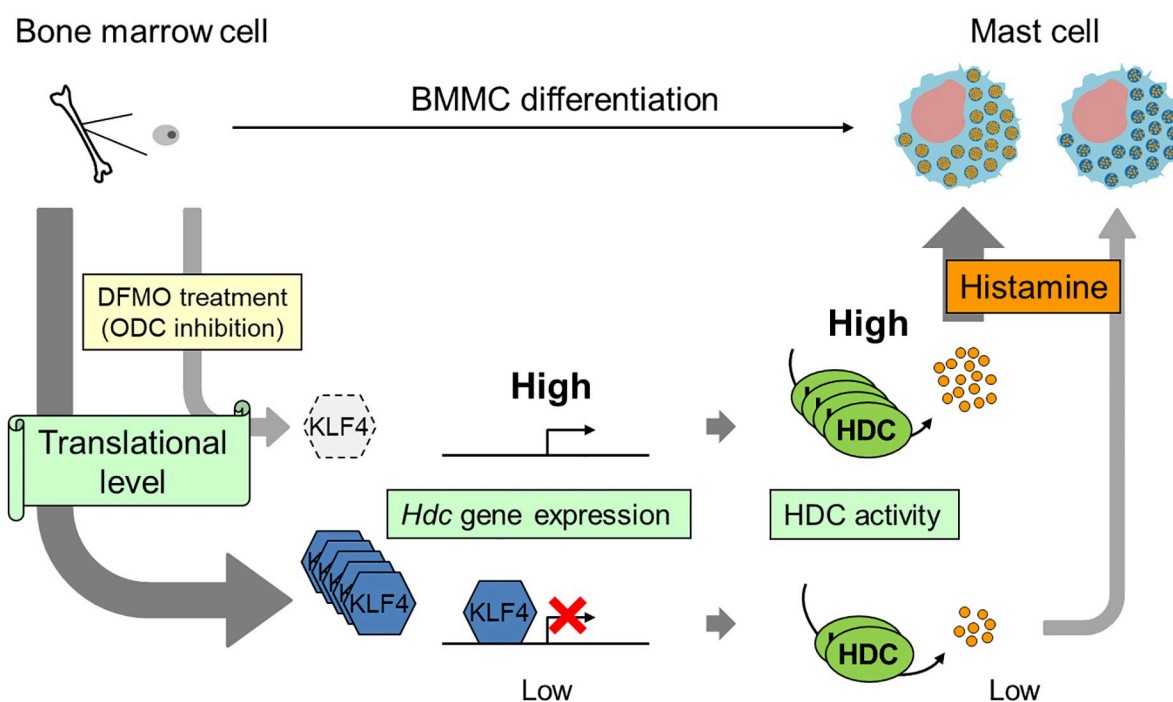

**Fig 7. Model of suppression of histamine synthesis by polyamines during BMMC differentiation.** The regulation of KLF4 synthesis depends on cellular polyamines at the translational level. Upon polyamine depletion, an upregulation of *Hdc* expression is caused due to the disappearance of KLF4. Higher levels of histamine is synthesized during differentiation.

secretary granules and degranulation response (Fig 1B–1F). Our data also indicated the presence of an inverse relationship between histamine and polyamine metabolism. A point of difference was that the intracellular histamine levels of our BMMCs increased more than 2-fold in response to DFMO treatment (Figs 2B and 3 and S1B Fig), although intracellular histamine levels of their BMMCs did not increase [17, 27]. This may be due to the difference in intracellular spermidine levels. It is known that only spermidine is required as a substrate for posttranslational modification of the eukaryotic translation initiation factor 5A (eIF5A). This modification, hypusination, is a two-step enzymatic reaction that only occurs in eIF5A, which is essential for cell proliferation and life [30–32]. A previous study indicated that a dramatic decrease of intracellular spermidine (below 5 nmol/mg protein) caused inhibition of hypusination [33]. The spermidine concentration was halved in our DFMO-treated BMMCs on day 21, however the level in their BMMCs was significantly reduced to 1.87 nmol/mg protein [17]. Therefore, inhibition of eIF5A may affect protein synthesis involved in histamine storage in granules. Recently, Carlos Acosta-Andrade *et al* reported that slowly decreasing intracellular polyamine levels in *Azin2* hypomorphic BMMCs showed an increase in histamine levels and HDC activity [34]. Our hypothesis is supported by this report.

An abnormal morphology of secretory granules in BMMCs was observed in response to treatment with low concentrations of DFMO. Serglycin (SG) exists as a core protein of proteoglycans in mast cell granules [12]. As seen in transmission electron micrographs of BMMCs of SG$^{-/-}$ mice, dense core formation was defective in SG$^{-/-}$ mast cell granules [35]. This was similar to the morphological changes observed in granules of DFMO-treated BMMC. It is known that the SG in BMMCs mainly attaches chondroitin sulfate. We suggest that morphologic abnormality of secretory granules by DFMO treatment may involve a decrease in SG and chondroitin sulfate due to polyamine depletion.

Here we showed that polyamine depletion maintains high HDC activity at the transcriptional level and increases of intracellular histamine during BMMC differentiation (Figs 2B, 4A and 4B). Previously, KLF4 has been reported to suppresses expression of *Hdc* in a gastric cancer cell [18]; we found that it is also involved in differentiation. The level of KLF4 decreased, without any change in mRNA levels, in response to polyamine depletion (Fig 4C and 4D). We classified the *Klf4* as a member of the 'polyamine modulon'. *Klf4* has a long GC-rich 5′-UTR, one of the characteristics of genes regulated by polyamines at the translational level. Several mechanisms of polyamine mediated stimulation of translation have been reported, such as ribosome shunting on the 5′-UTR [8], distant positioning of the CR sequence from the AUG [29], frameshifting at the termination codon (UGA) on the ORF [36], and miRNA mediated suppression of initiation codon recognition [37]. In this study, to clarify the mechanism of translational stimulation of KLF4 by polyamines, we used luciferase reporter plasmids containing several 5′-UTR versions of *Klf4* mRNA. First, *Klf4* had two initiation codons in the same frame, however, the long form from 1$^{st}$ ATG on E1 was untranslated, and only the short form from 2$^{nd}$ ATG on E2 was translated (Fig 5C). It has been reported that KLF4 N-terminal variance modulates reprogramming efficiency of induced pluripotent stem (iPS) cells [38]. These differences between long and short forms may depend on different tissue and cell types. Next, since the 5′-UTR on *Klf4* mRNA significantly inhibited luciferase activity in response to polyamine depletion, we focused on the uORF of *Klf4* mRNA. Upstream ORFs abound in mammalian mRNAs, their uORFs mostly suppress the translational efficiency of the downstream ORF [39]. *Amd1* mRNA, which encodes the polyamine biosynthesis rate-limiting enzyme AdoMetDC, has a unique uORF encoding MAGDIS, and polyamine depletion causes uORF-dependent translation stimulation [40]. Although the uORF of *Klf4* mRNA is 59 codons longer than MAGDIS, analysis of an uORF mutant of the reporter plasmid uORFmut did not detect translation stimulation by polyamines (Fig 5C). Recently, it has been reported that polyamines stimulate CHSY1 synthesis through the unfolding of the RNA G-quadruplex at the 5′-untrasnslated region [41]. A similar 12-nucleotide guanine quartet $(CGG)_4$ motif that can form RNA G-quadruplex structures is also present in the 5′-UTR of *Klf4* mRNA. Therefore, we hypothesize that this $(CGG)_4$ motif plays a key role in translational regulation by polyamines. However, enrichment of the $(CGG)_4$ motif in the 5′-UTRs of eIF4A-dependent mRNAs has been reported to be due to the formation of stable hairpin structures rather than G-quadruplexes [42, 43]. It is controversial whether RNA G-quadruplex structures are actually formed in cells.

Although *Klf4* is important as one of the Yamanaka factors of iPS cells [44], it also plays a role as a zinc finger type transcription factor that is involved in various processes of cell proliferation and differentiation [45]. Our finding, that translational regulation of KLF4 is mediated by polyamines, may represent a common mechanism regulating various processes occurring in *Klf4*-expressing cells.

## Supporting information

**S1 Fig. Original blot data_KLF4.**
(PDF)

**S2 Fig. Original blot data_Actin.**
(PDF)

**S3 Fig. Chromatin immunoprecipitation assay during BMMC differentiation in the absence or presence of DFMO.**
(PDF)

**S4 Fig. Cell number and intracellular amines in BMMCs in the absence or presence of DFMO and putrescine.**
(PDF)

## Acknowledgments

We thank Dr. A. J. Michael for his help in preparing this manuscript. We also thank Aventis Pharma and Torii Pharmaceutical Co. for providing DFMO and FUT-175, respectively.

## Author Contributions

**Conceptualization:** Kazuhiro Nishimura.

**Data curation:** Kazuhiro Nishimura.

**Formal analysis:** Kazuhiro Nishimura.

**Funding acquisition:** Kazuhiro Nishimura.

**Investigation:** Kazuhiro Nishimura, Moemi Okamoto, Rina Shibue, Toshio Mizuta, Toru Shibayama, Tetsuhiko Yoshino, Teruki Murakami, Masashi Yamaguchi, Satoshi Tanaka, Toshihiko Toida.

**Methodology:** Kazuhiro Nishimura, Satoshi Tanaka.

**Project administration:** Kazuhiro Nishimura.

**Resources:** Kazuhiro Nishimura.

**Supervision:** Kazuhiro Nishimura, Toshihiko Toida, Kazuei Igarashi.

**Validation:** Kazuhiro Nishimura.

**Visualization:** Kazuhiro Nishimura.

**Writing – original draft:** Kazuhiro Nishimura.

**Writing – review & editing:** Kazuei Igarashi.

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
