## [Decision Letter · Decision Letter 0]

23 Dec 2019

PONE-D-19-32767

KLF4 Is Required for Suppression of Histamine Synthesis by Polyamines During Bone Marrow-derived Mast Cell Differentiation

PLOS ONE

Dear Dr. Nishimura,

Thank you for submitting your manuscript to PLOS ONE. After careful consideration, we feel that it has merit but does not fully meet PLOS ONE’s publication criteria as it currently stands. Therefore, we invite you to submit a revised version of the manuscript that addresses the points raised during the review process.

ACADEMIC EDITOR: Both the reviewers find the study is interesting, however, as you can find from a reviewer comments noted below,  the authors should address euthanasia procedure and IACUC approval of ethical issues raised appropriately. There also concerns in figures quality and statistical analysis of data. The manuscript should be proof read for English language. 

We would appreciate receiving your revised manuscript by Feb 04 2020 11:59PM. To enhance the reproducibility of your results, we recommend that if applicable you deposit your laboratory protocols in protocols.io, where a protocol can be assigned its own identifier (DOI) such that it can be cited independently in the future. For instructions see: http://journals.plos.org/plosone/s/submission-guidelines#loc-laboratory-protocols

We look forward to receiving your revised manuscript.

Kind regards,

Dr. Sakamuri V. Reddy

Academic Editor

PLOS ONE

Journal Requirements:

Reviewers' comments:

Reviewer's Responses to Questions

**Comments to the Author**

1. Is the manuscript technically sound, and do the data support the conclusions?

Reviewer #1: Yes

Reviewer #2: Yes

2. Has the statistical analysis been performed appropriately and rigorously? 

Reviewer #1: Yes

Reviewer #2: No

3. Have the authors made all data underlying the findings in their manuscript fully available?

Reviewer #1: Yes

Reviewer #2: Yes

4. Is the manuscript presented in an intelligible fashion and written in standard English?

Reviewer #1: Yes

Reviewer #2: No

5. Review Comments to the Author

Reviewer #1: Dr. Igarashi's group is an undoubted expert group in polyamine metabolism worldwide. The previous record of scientific production by this groups is admirable and always with its proper hallmark of quality and scientific rigor. In the present manuscript, this group explores interesting metabolic connections and cross-talks between polyamine and histamine metabolism during bone marrow-derived mast cell differentiation. The findings indicate that polyamines are key in the translational regulation of KLF4, a zinc-finger transcription factor that is involved in the regulation of several important physiological processes, including differentiation. The findings are relevant and deserve to be published.

Reviewer #2: Although the manuscript is interesting in the inhibitory effects of polyamines on histamine synthesis there are some serious problems. The authors should be addressing them.

1) It is the most serious problem that animals sacrificed under no anesthesia, indicated that the condition is animal abuse. If you did, the paper is refuse in Ethical guideline.

2) Abstract: The background is very unclear. And the proposal is not state. There is no statement about the relationship between histamine and polyamine.

3) Abstract p2, line 28: What is cell function?

4) The authors focused some kind of amines. However, the authors speculate which amine is an important functional role in mast cell differentiation.

5) Results: The response and time course treated with DFMO were different in each amine. What did the authors speculate?

6) Results: p10, line 239-240: What did the results mean?

7) Figure1A: The authors should perform the significant test.

8) Figure 4C: The bands in beta-actin as internal control were too faint and no constant. The authors should improve them.

9) Figures: The marks such circles and squares in line graph were too big to be superimposed them, it is unknown the correct value and SD in all line graph. The authors should use a bar graph.

10) Discussion: The transcriptional activity in KLF4 generally confused the release of amines in mast cells. The authors should explain the relationships among DFMO, KLF4, and histamine in detail.

11) English sentences are too poor. English proofreading is required.

6. PLOS authors have the option to publish the peer review history of their article (what does this mean?). If published, this will include your full peer review and any attached files.

Reviewer #1: Yes: Miguel Ángel Medina

Reviewer #2: No

---

## [Author Response · Author response to Decision Letter 0]

4 Feb 2020

Dear Dr. Sakamuri V. Reddy:

We have now resubmitted our manuscript entitled “KLF4 is required for suppression of histamine synthesis by polyamines during bone marrow-derived mast cell differentiation” for publication. This revision contains modified data from new experiments performed according to the suggestions of the reviewers. We describe below our response to the Journal Requirements and the reviewers.

Response to Journal Requirements:

1: We prepared the revision following PLOS ONE's style requirements.

2: We provided the original blot data in Supporting information.

3: We deleted the phrase “data not shown” in our manuscript.

4: We added the Supporting Information in our manuscript.

Response to Reviewer #2:

 We thank Reviewer #2 for important comments. In this revision, we performed experiments to address the Reviewer #2’s concerns. The details are described below:

1): We described that mice were euthanized by cervical dislocation, during isoflurane anesthesia in p6, line 77.

2 & 3): We described about “cell function” and “relationships between histamine and polyamines” in Abstract p2, line 3-6. Putrescine and histamine are produced from ornithine and histidine by decarboxylation of amino acids, respectively. We suggested that similar chemical structure as biogenic amine may affect behavior to variable cellular processes.

4): In Fig 1, we showed that decreased polyamines by DFMO unaffected to BMMC differentiation because of no difference in degranulation results. The DFMO treatment occurred the decrease of cell numbers and the increase of histamine levels after BMMC differentiation. It was reported that endogenous histamine is important for mast cell differentiation by using HDC-/- mice; however, it did not deplete under our experimental condition.

5): We described in INTRODUCTION that cellular polyamines are stringently controlled by biosynthesis, degradation, and transport. Therefore, we speculated that the DFMO treatment usually occurs to decrease the levels of putrescine and spermidine, and spermine levels maintain by feedback regulation including the synthesis from spermidine and inhibition of degradation to spermidine. These phenomena have been reported in various cell lines.

6): It means that we could not reproduce under the similar condition of the previous report. The reason is unknown, we cannot investigate the moreover. However, we obtained similar phenomena with lower DFMO treatment.

7): We performed a statistical analysis in Fig 1A.

8): We changed western blotting data in Fig 4C.

9): We changed to be small the size of marks in Fig 2 and Fig 3.

10): Previous report and this study showed that transcription factor KLF4 suppresses expression of Hdc gene by binding on the promoter region. Therefore, the lower KLF4 protein by DFMO treatment maintains high HDC activities at the transcriptional level and increases of intracellular histamine. We explain about these relationships in DISCUSSION p23, line 400-403.

11): We performed English proofreading by Editage (www.editage.com).

Sincerely yours,

Kazuhiro Nishimura, Ph. D.

---

## [Decision Letter · Decision Letter 1]

13 Feb 2020

KLF4 is required for suppression of histamine synthesis by polyamines during bone marrow-derived mast cell differentiation

PONE-D-19-32767R1

Dear Dr. Nishimura,

We are pleased to inform you that your manuscript has been judged scientifically suitable for publication and will be formally accepted for publication once it complies with all outstanding technical requirements.

With kind regards,

Dr. Sakamuri V. Reddy

Academic Editor

PLOS ONE

Additional Editor Comments (optional):

Reviewers' comments:

Reviewer's Responses to Questions

**Comments to the Author**

1. If the authors have adequately addressed your comments raised in a previous round of review and you feel that this manuscript is now acceptable for publication, you may indicate that here to bypass the “Comments to the Author” section, enter your conflict of interest statement in the “Confidential to Editor” section, and submit your "Accept" recommendation.

Reviewer #2: All comments have been addressed

2. Is the manuscript technically sound, and do the data support the conclusions?

Reviewer #2: Yes

3. Has the statistical analysis been performed appropriately and rigorously? 

Reviewer #2: Yes

4. Have the authors made all data underlying the findings in their manuscript fully available?

Reviewer #2: Yes

5. Is the manuscript presented in an intelligible fashion and written in standard English?

Reviewer #2: Yes

6. Review Comments to the Author

Reviewer #2: The revised manuscript has significantly improved by adequately addressing the points made by the reviewers.

The reviewer recommend it for publication in your journal.

7. PLOS authors have the option to publish the peer review history of their article (what does this mean?). If published, this will include your full peer review and any attached files.

Reviewer #2: No

---

## [Editor Report · Acceptance letter]

18 Feb 2020

PONE-D-19-32767R1 

KLF4 is required for suppression of histamine synthesis by polyamines during bone marrow-derived mast cell differentiation 

Dear Dr. Nishimura:

I am pleased to inform you that your manuscript has been deemed suitable for publication in PLOS ONE. Congratulations! Your manuscript is now with our production department. 

With kind regards,

on behalf of

Dr. Sakamuri V. Reddy 

Academic Editor

PLOS ONE